# The Relationship between a Growth Mindset and Junior High School Students’ Meaning in Life: A Serial Mediation Model

**DOI:** 10.3390/bs13020189

**Published:** 2023-02-20

**Authors:** Hui Zhao, Ming Zhang, Yifei Li, Zhenzhen Wang

**Affiliations:** Faculty of Education, Henan Normal University, Xinxiang 453007, China

**Keywords:** junior high school students, growth mindset, meaning in life, psychological capital, core self-evaluations, serial mediation

## Abstract

A growth mindset is an individual’s belief that human intelligence can be changed through continuous practice and effort. The meaning in life signifies that individuals understand or see the meaning of their own life and are aware of their own goals and the values of their own life. Previous studies have shown that a growth mindset positively promotes individual emotional health and life happiness, but its relationship with meaning in life needs to be clarified. In this study, taking the self-determination theory and the broaden-and-build theory of positive emotions as a basis, we constructed a serial mediation effect model to test the mechanism of psychological capital and core self-evaluation in the relationship between a growth mindset and the meaning in life. A total of 565 students from Chinese junior middle schools participated in this study. The growth mindset, meaning in life, psychological capital, and core self-evaluation scales were used to collect the data for the study. The results indicated the following: (1) meaning in life was significantly predicted by growth mindset (*β* = 0.181, *p* < 0.001); (2) psychological capital and core self-evaluations played a mediating role in the influence of growth mindset on junior high school students’ meaning in life. The mediating role includes three mediating paths: the individual mediating effects of psychological capital, the individual mediating effects of core self-evaluations, the serial mediating effects of psychological capital and core self-evaluations. The results of this study confirm the benefits of a growth mindset, as well as the potential mechanism by which they impact meaning in life, which positively impacts junior high school students meaning in life.

## 1. Introduction

Mental health has always been a hot topic of general concern for scholars. Previous research found that meaning in life and mental health are closely related [1,2]. As a positive predictor of individual mental health, the basic connotation is that people are conscious of their own life objectives, tasks, and missions and understand, grasp, or see the significance of their own lives [3]. Individuals with high levels of meaning in life are more likely to behave altruistic and perceive higher life satisfaction, which relieves individual psychological pressure [4,5]. At present, the research on the acquisition and promotion mechanism of meaning in life mostly focuses on social relations, such as family intimacy, social support, and peer relations [6,7]. Positive or negative emotions include anxiety, stress, depression, and gratitude [8]. Personality traits include a personality hardiness and a big five personality [9,10]. However, despite understanding the importance of meaning in one’s life, only a few studies have examined the cognitive factors that affect it. Therefore, by exploring the cognitive influences on meaning in life, this study explains how the underlying mechanisms influence it. It also has an important value for strengthening individuals’ meaning in life.

The positive psychology field considers a growth mindset an important topic, defined as having a basic belief in one’s intelligence or ability. Individuals with this mindset hold the ‘growth ability view’, believing that intelligence has the characteristics of growth, shaping and regulation and can be continuously improved through dedicated study and training [11,12]. They tend to have more positive cognitive, emotional, and physiological reactions in the face of sudden changes [13]. The self-determination theory claims that everyone needs self-development. When individuals can flexibly control the interaction between themselves and the environment, they tend to self-innovate and seek interesting and challenging activities; they also take responsibility in the activities and find it easy to achieve their goals and values [14]. The broaden-and-build theory of positive emotions posits that positive emotional experiences reflect the positive feelings of the individual, are conducive to their growth and development, and have long-term adaptation value [15]. Some scholars have demonstrated that a growth mindset promotes aspects such as self-efficacy, well-being on a subjective level, emotional health, and life satisfaction, and other aspects [16,17]. How it relates to meaning in life, however, needs to be further investigated.

A critical period in a student’s identity development occurs during junior high school. During this period, they are vulnerable to the interference of bad emotions and problem behaviours as they attempt to establish their life’s purpose and meaning. Meng et al. investigated 5246 middle school students nationwide and found that 31.7% lacked meaning in their lives [18]. This is due to reasons such as the pressure of high intensity studies leading them to become bored, indifferent, and unmotivated, which then makes it difficult for them to experience and develop meaning in life [19]. Therefore, it is important to explore the direct correlation between junior high students’ growth mindset and their sense of meaning in life. The meaning-making theory holds that the psychological mechanism of individuals to obtain meaning in life varies in different situations [20]. Thus, it is also essential to explore the mechanism through which a growth mindset affects meaning in life.

The premise of this study is based on previous research and uses the self-determination theory and the broaden-and-build theory of positive emotions. This paper deeply explores the positive significance of the growth mindset and its potential mechanism through which it affects one’s meaning in life with junior high school students as the research object. The aim of this study is to enrich and expand the empirical research on the growth mindset and meaning in life. The concept of a growth mindset will be explored at a deeper level to understand whether it can enhance individual psychological capital leading to improved self-evaluation, thereby increasing one’s meaning of life. This research is conducted within the context of Chinese culture, with Chinese junior high school students as participants. It provides value for future research on meaning in life under other cultural contexts. Furthermore, it provides targeted suggestions and support for the prevention, intervention, and counselling of the meaning in life among junior middle school students.

## 2. Literature Review

### 2.1. The Relationship between Growth Mindset and Meaning in Life

As an important factor in an individual’s development and growth, the growth mindset is the belief that a person can improve intelligence, ability, morality, or personality. Such individuals view difficulties and failures from a more positive perspective and can effectively cope with helpless reactions such as anxiety, depression, and challenge avoidance [21,22]. The meaning in life is one in which individuals strive to achieve their goals and feel valued by others. As a cognitive emotion, it is generated through the subjective judgment of individuals [23]. When individuals are affected by negative emotions and are in a state of anxiety for an extended period of time, their perception of the meaning in life will be reduced [24]. As a result, cognition or thinking may affect the meaning in an individual’s life. According to the self-determination theory, humans tend to have positive growth and self-regulation. Individuals achieve self-realisation through the integration of goals and motivations, which can promote personality improvement and growth of mind [25] and then understand their true purpose in life. Previous empirical results suggest that a growth mindset is a significant indicator of people’s lives. When individuals believe in growth ability, they often have a positive life experience, thus generating high satisfaction [26]. Additionally, as a positive belief or cognition, a growth mindset focuses on self-growth and development [27,28]. When it focuses on the positive role of self-growth in daily life rather than passive defence, it is easier to obtain the meaning and value of life [29]. According to some theories, a growth mindset is a positive belief that enables individuals to achieve a sense of meaning in life. Guan proposed that contemporary youth suffer from a serious “hollow disease”; that is, they are pessimistic and lack a sense of existence and cognitive value. The core reason for this is the lack of life’s meaning perceivable. The first solution strategy is to change individuals thinking and cognition [30]. The COVID-19 pandemic and the long isolation period had consequences on the mental state of adults as well as the youth, it led to the ‘hollow disease’ gradually affecting younger individuals, with recent studies reporting that many middle school students hold an ‘indifferent’ negative attitude [31]. Thus, the research sample selected for this study consisted of junior high school students. A growth mindset is examined in relation to meaning of life and we put forward the following:

**Hypothesis** **1.**
*Growth mindset has a positive impact on a person’s meaning in life.*


### 2.2. Mediating Effect of Psychological Capital

According to Luthans, positive psychological capital refers to a healthy mental state that individuals exhibit as they grow and develop. Including resilience, optimism, self-efficacy, and hope [32]. The causal orientation theory states that individuals with autonomous orientation often rely on the value of self-recognition, and this positively affects their emotional state and behaviour in activities [33]. A growth mindset is often characterised by “actively seeking resources”, “focusing on the internal growth of individuals”, and being “willing to try different challenges” [34]. They often regard different tasks as an opportunity for self-growth so that they can be fully honed when facing problems, thereby improving their psychological resilience. Empirical research also found that individuals with a growth mindset can show better adaptability and handling ability when facing setbacks or difficulties. A growth mindset has been shown to play a positive role in education. When individuals have a higher level of growth mindset, they will be optimistic about the future and have a higher degree of academic self-recognition. Therefore, they will believe in their ability to deal with different academic problems, namely, high academic efficacy, which is an extension of self-efficacy. To some extent, it indicates that the growth mindset may be related to the dimension of efficacy in psychological capital [17,35]. Other studies found that mothers who raise children with a growth mindset show a high sense of hope, which is more conducive to children’s happiness [36]. Thus, a growth mindset and psychological capital exhibit a positive relationship. Moreover, psychological capital may also be positively influenced by a growth mindset.

Since the advent of positive psychology, the positive impact of psychological capital on an individual’s standard of living, growth, and development is becoming increasingly recognised by researchers. The broaden-and-build theory of positive emotions states is as follows: a positive feeling individual can expand their immediate thinking and actions and then build long-lasting individual resources, thus resulting in long-term adaptive benefits for the individual [37]. Psychological capital was found to be closely related to life in previous empirical studies conducted by the author. For example, a person’s psychological capital level is positively correlated with their level of life satisfaction [38,39,40]. When individuals are more satisfied with their living standards, they can pursue and have meaning in life [41]. Additionally, if individuals hold confidence, optimism, and other positive attitudes or emotions in the process of growth and development, they will be able to adapt well to the new environment and life and be full of hope for the future. These individuals are more likely to actively discover what life is all about, including its worth and goals [39,42]. Therefore, a person’s meaning in life may be affected by psychological capital. To sum up the above content, the study proposes the following:

**Hypothesis** **2.**
*The relationship between a growth mindset and meaning in life can be mediated by psychological capital.*


### 2.3. Mediating Effect of Core Self-Evaluations

Core self-evaluation is an assessment of a person’s cognitive and comprehensive abilities, which is the basis of other evaluations related to specific situations [43]. According to the theory of thinking mode, cognitive thinking is conducive to helping students build a coherent meaning system, thereby affecting individual goal setting, attribution, and belief in the effort. As a positive thinking mode, growth mindset tends to support a positive and optimistic attitude, more independent motives in action and confidence in one’s abilities, thus generating a positive self-evaluation [44]. Previous empirical studies show a negative relationship between low growth mindset and core self-evaluation. In a family background where parents have a low growth mindset, teenagers often cannot look at themselves positively and may have a low self-evaluation [45]. Kyoung et al. found that self-esteem is associated with a growth mindset in a study with 467 middle school students [46]. In terms of self-evaluation, self-esteem is one of the key components. It can be inferred that adolescents with positive thinking styles may have higher self-evaluation, which means that it is plausible that a growth mindset can positively contribute to core self-evaluation.

Individuals holding a high opinion of themselves tend to have a positive mental state and clear goals in life [47,48]. Perhaps it has to do with meaning in life. Consistent with the functional core self-evaluative mechanism summarised by Li, the schema mechanism explains the positive prediction of the meaning in life by core self-evaluation. When an individual forms a positive self-schema in their mind, it affects the subsequent emotional experience and behaviour activities [49]. Therefore, individuals with low core self-evaluation may have negative self-cognition, hold negative attitudes toward their conditions and the external environment, and easily lose their pursuit of life goals [50]. As a predictive factor of meaning in life, according to previous studies, a person with a positive core self-evaluation is likely to rate themselves positively in terms of their attitudes, and they will think that they can control their lives, thus becoming more capable and valuable [51]. Sun et al. also found through questionnaires that when individuals have a good self-evaluation, they will adopt a positive coping style when facing problems, which is conducive to the ownership and pursuit of life’s meaning [52]. The meaning in life is positively influenced by core self-evaluation. Considering the discussion above, this study proposes the following:

**Hypothesis** **3.**
*The relationship between a growth mindset and meaning in life can be mediated by psychological capital core self-evaluation.*


### 2.4. Serial Mediating role of Psychological Capital and Core Self-Evaluations in the Relationship between Growth Mindset and Meaning in Life

It is also important to explore the relationship between psychological capital and core self-evaluation. When individuals evaluate themselves, they are affected by positive psychological development. When facing difficulties, individuals with more positive psychological resources can recover more quickly and then generate positive incentives for themselves, ultimately forming a higher self-evaluation [53,54]. Research indicates that a high growth mindset is more likely than a low growth mindset to make people believe they can adapt to and dominate their behaviours and activities. When facing difficulties or setbacks, they tend to be more optimistic; that is, they have more positive psychological resources. Furthermore, positive self-awareness is associated with high core self-evaluations, high recognition of abilities, and better use of advantages [55]. They find it easier to achieve goals in daily life and experience life value and significance [56]. Therefore, a growth mindset may indirectly affect individuals meaning in life through psychological capital and core self-evaluation. Considering what has been discussed above, this study proposes the following:

**Hypothesis** **4.**
*A growth mindset can affect meaning in life through the serial mediating roles of psychological capital and core self-evaluation.*


This study is based on the self-determination theory as well as the broaden-and-build theory of positive emotions into consideration, using students in Chinese junior high schools as the research subjects, developing a serial mediation model based on growth mindset, meaning in life, psychological capital, and core self-evaluation (Figure 1). We explore how a growth mindset affects meaning in life and provide a theoretical framework for explaining its effects on the meaning in the lives of junior high school students and their educational interventions.

## 3. Materials and Methods

### 3.1. Participants

Using cluster sampling and taking a class as a unit, 620 junior middle school students in Henan Province, central China, were selected to participate in this study. A total of 565 questionnaires were collected, and the questionnaire completion rate was 91%. There were 264 boys (47.2%), 301 girls (52.8%), 153 seventh-grade students (27.8%, 73 boys, 80 girls), 229 eighth-grade-students (40.8%, 115 boys, 114 girls), 183-ninth grade students (31.4%, 76 boys, 107 girls), and the average student age are 14.05 years old. Informed written consent was obtained from the participants. The study was approved by the Ethics Committee of the Henan Normal University.

### 3.2. Measures

#### 3.2.1. Growth Mindset

Dweck’s growth mindset scale was used in this study (1999), which has eight items; the score includes four items that are positive (such as “You can always substantially change how intelligent you are”) and four reverse scoring items (such as “Your intelligence is something very basic about you that you can’t change very much”) [57]. We used a Likert scale of seven points to score the study; the range is from “completely disagree” to “completely agree”. During data analysis, we calculate the average score of the eight items. The higher the scores, the more growth mindset an individual has. There is good reliability and validity of the scale in China [58]. The present study found acceptable reliability values for the scale (α = 0.716).

#### 3.2.2. Meaning in Life

Wang’s meaning in life scale (MLQ) was used in our study (2013), which has 10 items, including 9 positive score questions (such as “I’m looking for my purpose and mission in life”) and 1 negative score question (such as “My life has no clear purpose”) [59]. A Likert scale of seven points was used to score the variable in this study from “completely disagree” to “completely agree”. During data analysis, calculate the average score of the 10 items. Scores that are higher, the greater the meaning in life. There is good reliability and validity of the scale in China [60]. The present study found acceptable reliability values for the scale (α = 0.757).

#### 3.2.3. Compound Psychological Capital

Lorenz’s compound psychological capital scale (CPC) was used in our study (2016), which has 12 positive scoring items (such as: “I am looking forward to the life ahead of me”) [61]. A Likert scale of seven points was used to score the study from “completely disagree” to “completely agree”. During data analysis, the average score of the 12 questions was calculated. A higher score indicates greater psychological capital. In China, the scale has a high level of reliability and validity [62]. This study found acceptable reliability values for the scale (α = 0.774).

#### 3.2.4. Core Self-Evaluation

Judge’s core self-evaluation scale (CSES) was used in our study (2003), which has 12 items, including 6 positive scoring questions (such as “I can succeed at the task”) and 6 negative scoring questions (such as “I am full of doubts about my ability”) [63]. A Likert scale of seven points was used to score the study from “completely disagree” to “completely agree”. During data analysis, calculate the average score of the 12 questions. Higher scores indicate more self-evaluations. In China, the scale has a high level of reliability and validity [64]. This study found acceptable reliability values for the scale (α = 0.707).

#### 3.2.5. Control Variables

According to previous studies, individuals of different genders and ages significantly differ in their perception and understanding of the meaning in life [65]. Therefore, in this research, age and gender were considered the main control variables.

### 3.3. Analytical Approach

In this study, a common method bias test was performed. A descriptive and correlational analysis was conducted using SPSS 26.0, and the serial mediation effect was examined using the PROCESS macro program developed by Hayes (2013) [66].

## 4. Results

### 4.1. Test of Common Method Biases

Since the statistical information is all from the questionnaire filled by the subjective intention of the participants, there are certain method errors. To avoid common method biases caused by the research method, the Harman single-factor analysis was used in our study [67]. Among them, the eigenvalues of nine factors were greater than one; for the first factor, the interpretation rate was 21.509%, which is below the critical standard of 40%, indicating that this study is within the acceptable range due to the influence of common method biases.

### 4.2. Descriptive and Correlational Analysis

In Table 1, the mean, standard deviation, and coefficient of correlation of the study variables are presented. According to the data, it was found that the four variables of growth mindset, meaning in life, psychological capital, and core self-evaluation showed positive correlations.

### 4.3. Regression Analysis

Our study used model 6 in process to conduct multiple regression analysis; that is, to analyse the impact of a growth mindset on life meaning and test the mediating role of psychological capital and core self-evaluation between a growth mindset and life meaning, and this method tested the integration of the serial mediation model [68]. As gender is a category variable, this study will transform it into a dummy variable: “male is the control group, 1 = female”. With gender and age as control variables. In Table 2, the specific results are as follows: positive predictive effects of growth mindset on psychological capital (*β* = 0.589, *p* < 0.001), psychological capital contributes significantly to meaning in life (*β* = 0.493, *p* < 0.001), and positive predictive effects of growth mindset were found on core self-evaluations (*β* = 0.198, *p* < 0.001); the impact of core self-evaluation on meaning in life was significantly positive (*β* = 0.131, *p* < 0.05).

### 4.4. Mediation Analysis

To determine whether mediation plays a significant role in this study, the Bootstrap method was used; the data was repeatedly sampled 5000 times repeatedly. Referring to Table 3, a growth mindset strongly influences meaning in life. Furthermore, the mediating effect of psychological capital (indirect effect 1), the mediation effect of the core self-evaluation (indirect effect 2), and the serial mediating effect of psychological capital and core self-evaluation (indirect effect 3) were significant. As a percentage of the total effect, 66.104% is accounted for by the mediation effect; psychological capital contributed 54.307% to the mediation effect. The mediating effect of core self-evaluations accounts for 4.869% of the effect, while the serial mediation effect of psychological capital and core self-evaluations is responsible for 6.929% of the effects. In detail, according to these results, among the three indirect effect paths in the overall effects, there were no zeros in the Bootstrap 95% confidence intervals, providing evidence that they were significant. Additionally, to determine whether there were significant differences between the indirect effects of different paths, they were also compared in pairs: In comparison 1, in terms of the confidence interval, there is no zero. The indirect effect 1, which differed from the indirect effect 2, clearly shows a significant difference. Applying the same idea to the comparison, it was found that there are two indirect effects: significant difference between effect 1 and effect 3. However, neither indirect effect 2 nor indirect effect 3 were significant. In Figure 2, you can see the specific model.

## 5. Discussion

Drawing on the theories of self-determination and broaden-and-build, this study examines the relationship between a growth mindset and meaning in life and its influencing path in a Chinese cultural context. The results show that in junior middle school, a growth mindset significantly influences individual meaning in life; a growth mindset can directly and significantly predict individual meaning in life. Mediation analysis shows that a growth mindset can strengthen junior high school students’ meaning in life by enhancing individual psychological capital and then improving individual core self-evaluation.

According to previous research, a growth mindset is closely connected to positive emotional experiences in individuals’ lives, such as life satisfaction and life happiness [69,70,71]. This study found that a growth mindset significantly predicts meaning in life; when junior high school students have a growth mindset, their meaning in life is higher. This result supports the previous scholars’ suggestion that a growth mindset, as an important positive predictor of individual growth, should inform future research on growth mindset and health [72,73]. At the same time, it also expands the theory of self-determination. Self-determination theory asserts that individuals are autonomous in their behaviour; they realise personal goals and values through their efforts, persistence, and strategies [74]. In contrast, individuals with a growth mindset strive to change and develop themselves through action; they show strong autonomy and choose active efforts or appropriate strategies to solve problems [75]. Therefore, such individuals have a strong sense of support and control, which is more conducive to understanding, pursuing, and realising individual goals. thus affecting an individual’s understanding of life’s meaning and value [76]. In the course of this study, it was determined that a growth mindset can positively and directly impact meaning in life in an individual.

According to the results of this study, psychological capital mediates the relationship between a growth mindset and meaning in life. The research has something in common with previous findings. A growth mindset can significantly predict psychological capital, discovered by Chen et al. [54], while Li found that psychological capital can significantly predict meaning in life [77]. The possible reasons are as follows. According to the attribution theory, the more inclined individuals are to a growth mindset, the more inclined they are to controllable factors rather than uncontrollable factors for difficulties and adversity, and the more they think they can overcome difficulties through effort [78]. A growth mindset also affects an individual’s positive emotions. A growth mindset may manifest a positive affective state during the development process and may be characterised by greater psychological resources, namely, psychological capital [79]. In the positive psychology perspective, the psychological capital is a positive, manageable psychological emotional state or psychological energy, which plays a positive role in individual attitudes [80]. When individuals have positive psychological resources such as resilience, hope, and efficacy, their mental health indicators will be higher. The more optimistic they are, the more they will be able to face life events and goals and gain a sense of successful experience and achievement in life so that they can own and perceive the meaning and value of life [81]. Therefore, through the mediating role of psychological capital, individuals with a growth mindset can indirectly affect meaning in life.

In the present study, core self-evaluations mediated the relationship between a growth mindset and meaning in life. According to Guo scholars, a growth mindset significantly predicts core self-evaluation [82]. Further, according to Xiang’s research, core self-evaluation was found to significantly predict meaning in life [83]. According to the theory of thinking mode, thinking mode plays a key role in setting goals, striving for beliefs, and reclusive ways. As a positive mode of thinking, a growth mindset can increase this ability. Individuals with this characteristic are good at using the perspective of development to look at themselves and believe they can achieve self-improvement through continuous effort [84]. When they encounter different situations, they adopt different strategies to actively cope with them and better regulate themselves. Therefore, their sense of control over the environment is strong and current tasks are not easily disturbed. They can make reasonable attribution according to the external environment so they can have a clear and high self-evaluation [85]. A positive attitude is also demonstrated by individuals with high core self-evaluations. They will think that they are capable, valuable, and in control. Moreover, they can experience more pleasure and satisfaction and less pressure and tension. They will perform better at work, be more successful in their career, and be more satisfied with work and life [86]. A growth mindset can be used as a protective factor to enable individuals to face the future life with a positive attitude, have a high self-evaluation, and finally realise the value and significance of life.

This study further found that psychological capital and core self-evaluation plays a serial mediation role between a growth mindset and meaning in life. This study found a significant positive correlation between psychological capital and core self-evaluation. This result is partly consistent with previous research [53,87]. People with higher levels of psychological capital have access to more psychological resources. They have more positive emotions and resilience to overcome difficulties and can look at themselves and the surrounding environment with an optimistic attitude. Hence, they have higher self-evaluation, which is significant to their development and improvement [88]. Self-determination theory argues that human beings are active creatures capable of realising and growing their potential. Each individual has a congenital, internal, and constructive tendency to develop himself and fully integrate with society [89]. This is in line with individuals who have a growth mindset. They focus on their growth, are more likely to accept challenging goals, and have stronger psychological resilience to solve problems in the face of difficulties [90]. When individuals have positive psychological resources, it helps them evaluate themselves with a positive attitude, enhance their sense of self-worth, form a perfect quality, and ultimately improve their meaning in life [91]. Therefore, if individuals want to own and pursue the value and significance in life, they should focus on shaping their growth-oriented thinking, leading them to have a positive outlook on the future, enriching their inner psychological resources, and attaching importance to forming positive self-identity towards themselves to achieve their life goals to the greatest extent.

This study examines the impact of a growth mindset on junior high school students’ meaning in life in Chinese culture. The results enrich the cross-cultural research fields of growth mindset, meaning in life, psychological capital, and core self-evaluation, which have important reference value for subsequent cross-cultural research.

Studies show that students from different cultural backgrounds have different self-reports on growth mindset, and culture affects the results of growth mindset, development, change, and the effect of the intervention [92]. This is consistent with the author’s conjecture. In addition, there is evidence to suggest that Chinese students are more likely than Americans to have a low-growth mindset [93]. At the same time, Steger scholars proposed that culture provides people with the conditions to obtain meaning from life. There are two dimensions to meaning in life: finding meaning and having meaning. As reported in a self-report study, individuals in the United States place a high priority on having meaning in life, whereas individuals in Japan place a high priority on pursuing meaning in life [94]. The above discussion provides explanations and explanations for the possible cross-cultural differences in this study.

### 5.1. Practical Implications

A critical period of self-development is occurring for middle-school students. The study results can also provide inspiration and reference for parents and educators to strengthen meaning in the life of middle school students, thereby supporting their mental health.

First, meaning in life is directly impacted by a growth mindset. Accordingly, during the junior high school years, teachers and parents need to recognise that meaning in life is important to foster students’ growth and development as healthy individuals. In the context of a meta-analysis of the research conducted by Zhang scholars, meaning in life appears to play a significant role in preventing mental illness among young people [95]. It can significantly negatively predict negative indicators of mental health (depression, anxiety, and other negative emotions). Therefore, educators and parents should change their traditional cognition and help students realise the positive impact of a high sense of life. On the other hand, targeted intervention strategies and programs should promote a growth mindset, such as curriculum training and group counselling intervention. Through regular training to improve students’ cognitive beliefs, previous studies show that a growth mindset intervention can significantly promote teenagers’ intellectual growth [96]. This can subsequently affect students’ recognition of their ability to strengthen the perception and pursuit of meaning in life.

Second, a growth mindset can enhance junior high school students’ psychological capital and affect their meaning in life. Based on cultivating a growth mindset, educators and parents should pay attention to the positive role of psychological capital. As a person’s potential capabilities and competitive advantage, psychological capital has the characteristics of investment and income [97]. Therefore, teenagers should learn to build positive internal psychological resources by themselves, improve their sense of self-efficacy through example demonstration, self-reinforcement, alternative reinforcement, and other ways so that individuals can experience the happiness of success in life and study, and awaken their own physiological and psychological potential needs [98]. In addition, young people should deliberately exercise their anti-quotient level, face difficulties with a positive attitude, gradually form an optimistic interpretation style, and feel hope and expectation for the future. At the same time, the school should rely on the class meeting course with the theme of mental health education, carry out psychological training and education according to the individual differences of students, help students to shape their psychological quality, and strengthen the bridge between growth mindset and meaning in life.

Third, a growth mindset can enhance the core self-evaluation of individuals, thereby affecting an individual’s meaning in life. Students should learn to find advantages, enhance their sense of identity, and then build good self-cognition. Secondly, educators should give students positive feedback and evaluation in their daily studies. Educators should avoid language violence toward students as it has negative impacts such as low self-esteem and low self-efficacy. Additionally, they should help students improve their self-identity as much as possible so that they can actively and comprehensively evaluate themselves. Finally, we should consider the core self-evaluation of parents. In previous studies, it has been shown that there is an inter-generational transmission of core self-evaluation between parents and their children during junior high school [99]. Therefore, parents, as the first teachers of their own children, should play a positive role in teaching by example.

### 5.2. Limitations and Direction for Future Research

This study has a number of limitations which should be noted. First, a cross-sectional study of questionnaires was conducted for this study, lacking the tracking study of data, and the results cannot determine the long-term effects among the four variables of growth mindset, meaning in life, psychological capital, and core self-evaluation. A longitudinal study of junior high school students may provide further insight into their meaning in life. Second, data were derived from the subjective self-reports of students, and the attitude or social approval effects when answering may affect the accuracy of the results. However, we informed students about the research purpose in advance during the test and prohibited mutual discussion while answering questions. The contents filled in are reasonable, and the order and number of questions are reasonably set to avoid this phenomenon. In the future, an individual’s growth mindset and meaning in life can be investigated through interviews and other methods.

Third, junior high school students are in an important stage of individual development, and their meaning in life is easily affected by other factors in daily life, such as social support and peer interaction. Previous studies show that interpersonal relationships can affect individuals’ pursuit and meaning in life to varying degrees [100]. Social support and peer interaction are subordinate to interpersonal relationships. Therefore, in the future, it could be included as a boundary condition in this model to further explore the impact of other factors on meaning in life and the psychological mechanism. Fourth, this study only included two mediation variables: psychological capital and core self-evaluation. However, previous studies show that individual motivation is crucial to the pursuit of meaning in life [101]; the theory of self-determination mentions that individual behaviour will be affected by internal motivation and external motivation, so motivation can be considered an intermediate variable for discussion in the future. Finally, this research sample was Chinese junior middle school students. Under different social and cultural (i.e., Western) backgrounds, individuals’ understandings of the growth mindset and life values may vary. In the future, we can examine whether a growth mindset can potentially enhance or suppress the positive impact of junior high school students’ meaning in life.

## 6. Conclusions

This study investigated whether a growth mindset is associated with having meaning in life through a questionnaire survey, as well as the underlying mechanisms of that relationship. The following conclusions are drawn: after controlling for the age and gender variables, a growth mindset can significantly predict meaning in life, core self-evaluation, and psychological capital. Additionally, a growth mindset can affect meaning in life through psychological capital and core self-evaluation, respectively, and can also affect the meaning in life by enhancing the individual psychological capital and then influencing individual core self-evaluation.

## Figures and Tables

**Figure 1 behavsci-13-00189-f001:**
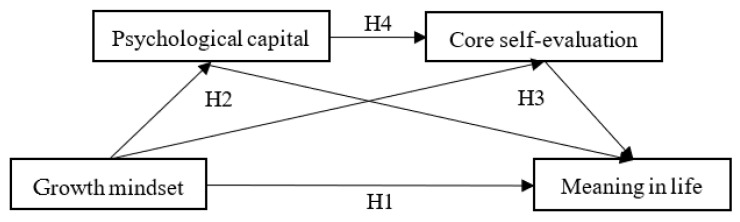
Theoretical research model.

**Figure 2 behavsci-13-00189-f002:**
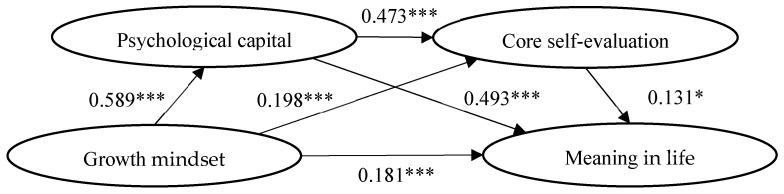
Roadmap of the influence of growth mindset on Meaning in Life. Note. * *p* < 0.05, *** *p* < 0.001.

**Table 1 behavsci-13-00189-t001:** The mean, standard deviation, and correlation of the study variables.

	M	SD	1	2	3	4	5	6
1. Gender	0.533	0.499	1					
2. Age	14.053	0.770	0.051	1				
3. Growth mindset	5.109	0.792	−0.057	−0.148 **	1			
4. Meaning in life	5.05	0.911	0.008	0.024	0.449 **	1		
5. Psychological capital	4.890	0.869	−0.099 *	−0.055	0.537 **	0.615 **	1	
6. Core self−evaluation	4.989	0.676	−0.052	−0.036	0.553 **	0.521 **	0.729 **	1

Note. N = 565, gender is a dummy variable: 0 = male; 1 = female, * *p* < 0.05, ** *p* < 0.01.

**Table 2 behavsci-13-00189-t002:** Regression analyses results.

Variable	Psychological Capital			Core Self−Evaluation			Meaning in Life		
	β	SE	t	β	SE	t	β	SE	t
Gender	−0.122	0.061	−1.973 *	0.026	0.038	0.707	0.117	0.059	1.980 *
Age	0.030	0.041	0.762	0.027	0.025	1.106	0.086	0.039	2.234 *
Growth mindset	0.589	0.039	14.953 ***	0.198	0.028	7.000 ***	0.181	0.046	3.882 ***
Psychological capital				0.473	0.025	18.452 ***	0.493	0.015	9.612 ***
Core self−evaluation							0.131	0.066	1.968 *
R^2^	0.294			0.570			0.413		
F	77.821			185.648			78.546		

Note. * *p* < 0.05, *** *p* < 0.001.

**Table 3 behavsci-13-00189-t003:** Summary of indirect effects using the bootstrapping method.

Path	Effect	Boot SE	Boot LLCI	Boot ULCL	Relative Mediation Effect
Direct effect	0.181	0.047	0.089	0.272	33.895%
Total indirect effect	0.353	0.033	0.289	0.420	66.104%
Ind1	0.290	0.036	0.221	0.361	54.307%
Ind2	0.026	0.014	0.001	0.056	4.869%
Ind3	0.037	0.018	0.001	0.074	6.929%
Compare 1(Ind1-Ind2)	0.264	0.045	0.176	0.354	
Compare 2(Ind1-Ind3)	0.254	0.048	0.158	0.351	
Compare 3(Ind2-Ind3)	−0.011	0.008	−0.031	0.001	

Note. Ind1 indicates the path: ‘Growth mindset → Psychological capital → Meaning in life’; Ind2 indicates the path: ‘Growth mindset → Core self-evaluation → Meaning in life’; Ind3 indicate the path: ‘Growth mindset → Psychological capital → Core self-evaluation → Meaning in life’.

## Data Availability

The datasets supporting the conclusions of this study are available from the corresponding author on reasonable request.

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
