# Peer review of "The Relationship between a Growth Mindset and Junior High School Students’ Meaning in Life: A Serial Mediation Model"

_behavsci, 2023, doi:10.3390/bs13020189_

Round 1

Reviewer 1 Report

The paper is well-written and analyses seem ok to me.

I have only minor comments, below details.

############################################################

ABSTRACT

############################################################

The authors could briefly describe what "meaning in life" and "growth mindset" mean

############################################################

INTRODUCTION

############################################################

The author(s) should justify why they focus on Junior High School Students (instead of normal population).

"In the 20th National Congress of the Chinese Communist Party"

Which year? Not crucial but would help contextualization

"Personality traits include a tough personality"

What is a thought personality?

"Based on the existing literature, to deeply explore the positive significance of the growth mindset and its potential mechanism affecting meaning in life, the Broaden-and- Build Theory of positive emotions is integrated with the Self-Determination Theory and uses participants who are junior high school students from China"

1) The Broaden-and-Build Theory is not defined.

2) this sentence should be unpacked. One does not take a theory, but *develop a design* based on such theories

3) *use participants* is weird (maybe describe your sample in another sentence)

############################################################

2.1. Relationship between growth mindset and meaning in life

############################################################

"post-epidemic era"

What is that? Covid?

############################################################

2.2. Mediating Effect of psychological capital

############################################################

"Luthans believed that"

Who is Luthans? Reference?

############################################################

2.3. Mediating Effect of core self-evaluations

############################################################

"Sun scholars"

?

############################################################

4.3. Regression Analysis

############################################################

"Our study utilises Model 6 to analyse"

What is model 6?

Author Response

Dear reviewers1, 

      Thanks for your valuable comments. These comments further stimulated our thinking and contributed to the improvement of the quality of this paper. We have revised our paper according to your comments. Please see the attachment.

Reviewer 2 Report

The article is relevant at both scientific and practical levels. The authors provide a theoretical justification and fulfil all the conditions for a scientific article. However, it is not clear whether the ethics of the investigation were respected. The authors of the article should be sure to describe this. It is also proposed that the first sentence of the first article be dropped as redundant ("In the 20th National Congress...). The list of references needs to be reviewed, as there are some inaccuracies (e.g. 88).

Author Response

Dear reviewers2, 

      Thanks for your valuable comments. These comments further stimulated our thinking and contributed to the improvement of the quality of this paper. We have revised our paper according to your comments. Please see the attachment.
